# Opportunities and Risks of Disaster Data from Social Media: A Systematic Review of Incident Information

Matti Wiegmann[1,2], Jens Kersten[2], Hansi Senaratne[3], Martin Potthast[4], Friederike Klan[2], and Benno Stein[1]

[1]Bauhaus-Universität Weimar
[2]German Aerospace Center (DLR), Institute of Data Science
[3]German Aerospace Center (DLR), German Remote Sensing Data Center
[4]Leipzig University

**Correspondence:** Matti Wiegmann (matti.wiegmann@uni-weimar.de)

**Abstract.** Compiling and disseminating information about incidents and disasters is key to disaster management and relief. But due to inherent limitations of the acquisition process, the required information is often incomplete or missing altogether. To fill these gaps, citizen observations spread through social media are widely considered to be a promising source of relevant information, and many studies propose new methods to tap this resource. Yet, the overarching question of whether, and under which circumstances, social media can supply relevant information (both qualitatively and quantitatively) still remains unanswered. To shed some light on this question, we review 37 disaster and incident databases covering 27 incident types, compile a unified overview of the contained data and its collection processes, and identify the missing or incomplete information. The resulting data collection reveals six major use cases for social media analysis in incident data collection: (1) impact assessment and verification of model predictions, (2) narrative generation, (3) recruiting citizen volunteers, (4) supporting weakly institutionalized areas, (5) narrowing surveillance areas, and (6) reporting triggers for periodical surveillance. Furthermore, we discuss the benefits and shortcomings of using social media data for closing information gaps related to incidents and disasters.

## 1 Introduction

A disaster is a hazardous incident, natural or man-made, which causes damage to vulnerable communities that lack sufficient coping and relief capabilities (Carter, 2008).[1] Key elements to disaster management are preparedness, early detection, and monitoring a disaster from its sudden, unexpected onset, to its unwinding, and its aftermath. Disaster-related data may be obtained from sensor telemetry, occurrence metadata, situation reports, and impact assessments. Various stakeholders benefit

---

[1]The International Federation of Red Cross (IFRC, 2017) provides a more detailed definition: "A disaster is a sudden, calamitous event that seriously disrupts the functioning of a community or society and causes human, material, and economic or environmental losses that exceed the community's or society's ability to cope using its resources. Though often caused by nature, disasters can have human origins."

from receiving such data, including task forces, relief organizations, policymakers, investors, and (re-)insurers. Not only data about ongoing incidents, but also about past ones is crucial to enable forecasting efforts, and to prepare for future incidents. The broad range of potential incidents and their ambient conditions require an equally broad range of monitoring techniques, each with their benefits and limitations: Remote-sensing provides spatial coverage, but is often heavily delayed and with low resolution; ground-sensors and scientific staff are fast and precise, but costly and far from ubiquitous; and citizen observers are ubiquitously available, but need training and incentivization to generate reliable observations. As a consequence, disaster monitoring is often spatially sparse and temporally offset, while underfunding causes further systematic information gaps.

A rising trend in the disaster relief community is to fill information gaps through citizen observations, ranging from the registration of tornado sightings and the verification of earthquake impact to reporting hail diameters and water levels. The traditional way of acquiring this information is to actively carry out surveys and to operate hotlines, requiring significant staff and a high level of engagement by citizens. In recent years, however, new information sources are increasingly being tapped: blogs, websites, news (Leetaru and Schrodt, 2013; Nugent et al., 2017), and "citizen sensors" on social media. The promise of passively collecting disaster-related information from social media has spawned pioneering research, from detecting earthquakes to estimating the impact of a flood. However, despite several statements of interest (GDACS, 2020) and early applications—like "Did You Feel It?" (DYFI) by the USGS (2020) to validate an earthquake's impact—most practical attempts to utilize disaster-related information from social media have yet to be acknowledged by professionals (Thomas et al., 2019).

Given the many approaches that have already been proposed to exploit citizen observations from social media for disaster-related tasks, it seems prudent to take inventory, and to refine our understanding of the information gaps that are supposed to be closed: (1) What information is missing in the current acquisition process and what information is too difficult or too expensive to acquire by relief organizations in a complete and consistent way? (2) Can we expect to find this information on social media? (3) What are the risks involved when social media is integrated as a source in the acquisition process? The paper in hand contributes to answering these questions by collating the information extraction from social media to date, and the observable gaps in the incident information collected by traditional means:

– We present a systematic survey of 37 disaster and incident databases, covering a broad range of disasters, hazardous incidents, regions, and timescales. We formalize the data acquisition process underlying the databases, which produces six relevant data points from any of three traditional information sources with a defined spatio-temporal resolution.

– We review the information gaps left open by the acquisition processes, i.e., their recall, by assessing the comprehensiveness of each database. Six major opportunities for social media-based citizen observations are identified in this respect: (1) impact assessment and model verification, (2) narrative generation, (3) recruiting citizen volunteers, (4) supporting weakly institutionalized areas, (5) narrowing surveillance areas, and (6) reporting triggers for periodical surveillance.

– To assess the risks of using social media data to fill the information gaps, we present an overview of the tradeoff between the limited recall of traditional information sources revealed by our survey and the limited precision of information extracted from social media based on clues given from past research in the field.

## 2 Related Work

Since the current landscape of disaster information systems has a variety of issues, there are also varied attempts at resolving them. Some organizations created curated collections of disasters to provide a unified index (ACDR, 2019) and harmonize disaster data (Below et al., 2010), to study disaster epidemiology (CRED, 2020), to cover new regions (La Red, 2019), or for profit (MunichRe, 2019; SwissRe, 2020; Ubyrisk Consultants, 2020). Other organizations started collaborations (GDACS, 2020), unified subordinates (NOAA, 2019; EU-JRC, 2020), or aggregate other resources (OCHA, 2019; RSOE, 2020). Even citizens contribute collaboratively through the recent disasters list by Wikimedians for Disaster Response (2017), the Wikinews (2020) collection on disasters, and the ongoing events and disaster categories of Wikipedia (2020).

Two recent meta-studies analyze the prerequisites of using social media for relief efforts by outlining the general patterns of social media usage during disasters: According to the first study by Eismann et al. (2016), the primary use case is always to acquire and redistribute factual information, followed by any one of five incident-specific secondary uses: (1) to disseminate information about relief efforts, fundraising activities, early warnings, and to raise awareness on natural disasters, (2) to evaluate preparedness for natural disasters and biological hazards, (3) to provide emotional support during natural disasters and societal incidents, (4) to discuss causes, consequences, and implications of biological hazards and technological and societal incidents, and (5) to connect with affected citizens during societal incidents. According to studies by Reuter et al. (2018) and Reuter and Kaufhold (2017), these usage patterns can be categorized in a sender-receiver-matrix, describing four communication channels: (1) information exchange between authorities and citizens, (2) self-help communities between citizens, (3) inter-organizational crisis management, and (4) evaluation of citizen-provided information by authorities. The operators of disaster information systems consider primarily the uni-directional channel of citizen-to-organization communication. One of those operators, the Global Disaster Alert and Coordination System (GDACS, 2020) of the United Nations and the European Commission remarks that the extraction of citizen observations is the key benefit of social media for their own, sensor-based information system, specifically regarding "assessing the impact of a disaster" on the population to extend and verify traditional models, and "assessing the effectiveness of response" including the extraction of secondary events like building collapses. In another survey, Reuter et al. (2016) assert that "the majority of emergency services have positive attitudes towards social media."

Most academic works since the pioneering publications by, for example, Palen and Liu (2007), conform with the assessment made by GDACS and focus on extracting information from citizen observations by studying how to infer influenza infection rates (Lampos and Cristianini, 2012), track secondary events (Chen and Terejanu, 2018; Cameron et al., 2012), estimate damages and casualties (Ashktorab et al., 2014), enhance the situational awareness of citizens (Vieweg et al., 2010), coordinate official and public relief efforts (Palen et al., 2010), disseminate information and refute rumors (Huang et al., 2015), generate summaries (Shapira et al., 2017), and create social cohesion via collaborative development (Alexander, 2014). Other research scrutinizes the problem of incident- or region-specific information systems by studying methods to detect earthquakes (Wald et al., 2013; Sakaki et al., 2010, 2013; Robinson et al., 2013; Flores et al., 2017; Poblete et al., 2018), wildfires, cyclones, and tsunamis (Klein et al., 2013) from Twitter streams, map citizen sensor signals to locate these incidents (Sakaki et al., 2013; Middleton et al., 2014), ingest disaster information systems for flash floods and civil unrest exclusively with social media data

(McCreadie et al., 2016), and explore the technical possibilities of combining social media streams with traditional information sources in tailored information systems (Thomas et al., 2019). A comprehensive survey of the academic work in crisis informatics has been presented by Imran et al. (2018). Despite significant prior work on techniques and algorithms to detect hazardous incidents from social media streams and to extract corresponding information, the majority of approaches only explore a narrow selection of disaster types, based on little systematic discussion of the needs of traditional disaster information systems, and ignoring the wealth of established remote sensing methods. As of yet, there is little understanding of the potential of social media in general, and whether computational approaches generalize to the full scope of hazardous incidents.

Several comprehensive monitoring systems have been proposed to generalize from studying particular events or focusing on a singular region or analysis method and to effectively expose disaster management to social media data. Twitcident (Abel et al., 2012) is a framework for filtering, searching, and analyzing crisis-related information that offers functionalities like incident detection, profiling, and iterative improvement of the situational information extraction. Data acquisition from Twitter based on keyword, and a human-in-the-loop tweet relevance classification and tagging have been implemented for the Artificial Intelligence for Disaster Response (AIDR) system (Imran et al., 2014). McCreadie et al. (2016) propose an Emergency Analysis Identification and Management System (EAIMS) to enable civil protection agencies to easily make use of social media. The system comprises a crawler, service, and user interface layer and enables real-time detection of emergency events, related information finding, and credibility analysis. Furthermore, machine learning is employed, trained with data gathered from past disasters to build effective models for identifying new events, tracking their development to support decision making at emergency response agencies. Similarly, the recently proposed Event Tracker (Thomas et al., 2019) aims at providing a unified view of an event, integrating information from news sources, emergency response officers, social media, and volunteers.

There is an apparent need to identify current information gaps and issues of operational disaster information systems, as well as to investigate the potential of utilizing social media data to fill these gaps and to augment traditionally used data sources, such as in-situ data, satellite imagery, and news feeds, with social media data. Recent research on event metadata extraction and management (McCreadie et al., 2016) forms a starting point for their integration into established disaster information systems.

## 3  Survey Method

A key prerequisite for an in-depth analysis of the gaps in incident information databases is a systematic review of the data that is currently collected across disaster types. In a first step, we narrowed the scope of disaster types to the set of most relevant ones, while maintaining diversity. We started with the de-facto-standard, top-down taxonomy used by EM-DAT (Below et al., 2009), which is based on work from GLIDE, DesInventar, NatCatSERVICE, and Sigma. It has also been closely adapted by the IRDR (2020) and appears to be more scientifically sound than, for example, the glossary of PreventionWeb, the typology of RSOE, and the bottom-up Wikipedia category graphs. We reduced the dimensionality of the type spectrum to a manageable degree by excluding exceedingly rare incident types (i.e., meteorite impacts) and combining types that are also commonly combined in the other databases (i.e., coastal and riverine floods) without crossing over sub-type hierarchies.

**Table 1.** List of disaster groups along corresponding disaster types and numbers of corresponding disasters in **EM-DAT**, **GLIDE**, **Wikipedia**, **Wikidata**, and in other **incident** databases since 2008. Unavailable or not applicable information is marked with **–**, and ∨ denotes that the disaster counts are added to the disaster in the next row, due to type subsumption.

| Disaster Group | Disaster Type | EM-DAT | GLIDE | Wikipedia | Wikidata | Incident DB | Source |
|---|---|---|---|---|---|---|---|
| Biological | Disease outbreak | 301 | 312 | – | 67 | 33,667 | CDC (2020); ECDC (2020) |
| Climatological | Drought | 182 | 94 | 37 | 27 | 29,922 | SWDI (2020); EDO (2020); NDMC (2020) |
| | Wildfire | 105 | 34 | 195 | 393 | 3,402 | GWIS (2020); SWDI (2020); GFW (2020) EFFIS (2020); NIFC (2020) |
| Geophysical | Earthquake | 273 | 196 | 1,147 | 1,950 | 1.6 mio | SWDI (2020); USGS (2020); IRIS (2020) |
| | Landslide (dry) | 6 | 94 | 117 | 78 | 6,789 | SWDI (2020); NASA (2020) |
| | Tsunami | 12 | 13 | 89 | 21 | 10,094 | NCTR (2020) |
| | Volcanic | 44 | 53 | 60 | 72 | 82 | NCEI-V (2020); BGS (2020); GVP (2020) |
| Hydrological | Landslide (wet) | 213 | 9 | 78 | 30 | 2,011 | SWDI (2020); ESSL (2020) |
| | Flood | 1680 | 848 | 169 | 196 | 61,558 | SWDI (2020); Brakenridge (2020) EFAS (2020); GLOFAS (2020) |
| Meteorological | Blizzard | 97 | 95 | 123 | 56 | 32,901 | SWDI (2020); ESSL (2020) |
| | Cold wave | 130 | – | 75 | 30 | 16,737 | SWDI (2020) |
| | Dust storm | 5 | – | 7 | 4 | 720 | SWDI (2020) |
| | Hail | 16 | – | 103 | 13 | 99,002 | SWDI (2020); ESSL (2020) |
| | Heat wave | 63 | 8 | 90 | 58 | 13,470 | SWDI (2020) |
| | Tornado | 56 | 24 | 295 | 123 | 19,847 | SPC (2019); ESSL (2020) MRCC (2017); THP (2020) |
| | Tropical storm | 615 | 410 | – | 30 | 19,253 | IBTrACS (2020); OCM (2020) |
| | Fog/Haze | 1 | – | – | 1 | 6528 | SWDI (2020) |
| | Thunderstorm | 132 | – | – | – | 145,470 | SWDI (2020) |
| | Rain | 1 | – | – | – | 13,230 | SWDI (2020) |
| | Wind | 101 | – | – | – | 37,671 | SWDI (2020) |
| Industrial | Chemical/Substance | 25 | ∨ | – | 81 | 8,655 | EFSA (2020) |
| | Radiation | 0 | ∨ | 51 | 34 | 1,173 | CNS (2020) |
| | Structure hazards | 239 | 47 | – | 300 | – | eMARS (2020) |
| Transportational | Aviation | 183 | ∨ | – | 2,165 | 26,059 | ICAO (2020) |
| | Railway | 98 | ∨ | – | 20 | 2,992 | ERAIL (2020) |
| | Maritime | 486 | ∨ | – | 49 | 2,336 | IMO (2020) |
| | Traffic | 764 | 154 | – | 66 | – | ITF (2020) |

Table 1 shows the resulting taxonomy of disasters, and the number of corresponding entries within EM-DAT and GLIDE as the largest expert-built disaster databases with global reach, as well as in Wikipedia and Wikidata, representing global bottom-up collaborative projects. The table also lists the existing incident databases and information systems of the major academic- and public institutions and NGOs for each disaster type and their cumulative number of entries in the time frame. Only disasters between 2008 and 2019 were counted, where social media started to be sufficiently widespread among the public, and since all surveyed databases had consistent coverage from 2008 onward. The table illustrates the differences in size between disasters recorded by experts in EM-DAT and Glide, by citizens in Wikipedia and Wikidata, and the notion of incidents in the other databases, where incident types are rarely systematically covered. In addition to the four mentioned disaster databases, we surveyed 33 further ones, and altogether 27 disaster types.

**Table 2.** Taxonomy of the information commonly collected about disasters.

| Dimension | Category | Definition | Examples |
|---|---|---|---|
| Data | Metadata | Structured data about an event | Date, time, location, disaster type, verification status, common name |
| | Sensory | Measured, type-specific information | Magnitude, depth, and severity |
| | Impact | Effects on the population | Damages caused, fatalities, injuries, displacements |
| | Causal relations | Causes and effects of the event | Trigger, follow-up |
| | Narrative | Detailed description of the event | Episode narrative, description |
| | Assessment | Reaction to the event | Response action taken, lessons learned |
| Source | Surveillance system | Automatic detection | Seismographs, buoys |
| | Expert | Assessment by trained persons | Meteorologists, park rangers |
| | Citizen observations | Observations by untrained persons | Call-ins, social media, newspaper |
| Resolution | Spatially dense | All areas are surveilled | Satellite imagery, weather stations |
| | Spatially constrained | Only relevant areas are surveilled | Plate boundaries, plane terminals |
| | Temporally periodical | Area is preemptively surveilled, without the need for a trigger | Seismograph, thermometer, buoys |

To overview the available incident data, we devise the taxonomy shown in Table 2, selected the largest database of each incident type as a representative, and judged the existence and completeness of each category in Table 3. The taxonomy organizes the relevant information within three dimensions relevant to our research questions: (1) The *data* collected for each incident type shows which information is in demand and which is difficult to acquire. (2) The *source* of the occurrence information and who detected the incident shows where citizen observations are meaningful and where surveillance systems or experts are preferable. (3) The spatial and temporal *resolution* shows the gaps in the acquisition process that can be filled by social media data. Dimensions beyond the scope of this survey include the typical presentation used for analysis, the involvement of post-processing and validation, and whether reports are qualitative or quantitative.

Based on the categories shown in Table 2, we examined the aforementioned databases regarding gaps in the collected data, by checking each database for the existence and completeness of information from each category. To acknowledge the diversity of disaster types, and to avoid exaggerated expectations, a category was rated "existent" if the database contained at least one piece of information from that category, and "incomplete" when less than 90% of the entries comprise the respective information. A source was rated "existent" if it contributes to the acquisition process, either with a reference to the source in the database, or by analyzing the database owner's description of the acquisition process. No sources were found "incomplete," but we noted the distribution of the participating sources whenever possible. Spatial resolution was rated "constrained" if only selected areas are surveyed (e.g., airports or forests), and "dense" otherwise. Temporal resolution was rated "periodical" if surveillance is scheduled in intervals instead of on-demand, and if it does not require a trigger event. The resolution was marked "incomplete" if the surveillance strategy does not fully cover the target, e.g., when some areas are not surveyed due to technical, jurisdictional, or financial constraints, and if incidents might be missed altogether.

**Table 3.** Assessment of the information collected in incident databases following our information taxonomy in Table 2. The **x** denotes existing, the **\*** incomplete information. The abbreviations correspond to the categories in the taxonomy. Data: **M**etadata, **S**ensor data, **I**mpact data, **R**elations, **N**arrative, **A**ssessments. Sources: **S**urveillance, **E**xperts, **C**itizens. Resolution: spatially dense (**S/D**) or constrained (**S/P**) and temporally periodical (**T/P**).

| Group | Type | Data | | | | | | Source | | | Resolution | | | Reference |
|---|---|---|---|---|---|---|---|---|---|---|---|---|---|---|
| | | M | S | I | R | N | A | S | E | C | S/D | S/P | T/P | |
| Biological | Disease outbreak | x | − | x | − | x | x | − | x | − | x* | − | − | CDC (2020) |
| Climatological | Drought | x | − | x* | x | x* | − | .73 | .26 | .01 | x* | − | x | SWDI (2020) |
| | Wildfire | x | − | x* | x | x | − | − | .83 | .17 | − | x* | − | SWDI |
| Geophysical | Earthquake | x | x | x* | x | x* | − | x | − | − | − | x | x | NCEI-EQ (2020) |
| | Landslide (dry) | x | − | x | x* | − | − | − | .69 | .31 | − | x | − | NASA (2020) |
| | Tsunami | x | x | x* | x | x* | − | .22 | .56 | .22 | − | x | x | NCEI-T (2020) |
| | Volcano | x | − | x* | x | x* | x* | x | x | − | − | x | x | NCEI-V (2020) |
| Hydrological | Landslide (wet) | x | − | x | x | x | − | .01 | .83 | .16 | − | x | − | SWDI |
| | Flood | x | − | x* | x | − | − | .31 | .51 | .18 | x* | − | x* | Brakenridge (2020) |
| Meteorological | Blizzard | x | − | x* | x | x | − | .46 | .38 | .16 | x | − | − | SWDI |
| | Cold wave | x | − | x | x | x | − | .31 | .51 | .18 | x* | − | x | SWDI |
| | Dust storm | x | − | x | − | x | − | .06 | .73 | .21 | x* | − | − | SWDI |
| | Hail | x | x | x | x | x | − | .02 | .51 | .47 | x* | − | x | SWDI |
| | Heat wave | x | − | x* | x | x | − | .83 | .06 | .11 | x* | − | x | SWDI |
| | Tornado | x | x | x | − | x | − | − | .86 | .14 | − | x* | − | SWDI |
| | Tropical storm | x | x | x* | − | x | − | .22 | .61 | .17 | x | − | x | SWDI |
| | Fog/Haze | x | − | x | x | x | − | .93 | .05 | .02 | x* | − | x | SWDI |
| | Thunderstorm | x | x | x* | − | x | − | .09 | .57 | .34 | x | − | x* | SWDI |
| | Rain | x | − | x* | − | x | − | .37 | .28 | .34 | x* | − | x | SWDI |
| | Wind | x | x | x* | − | x | − | .61 | .29 | .10 | x* | − | x | SWDI |
| Industrial | Chemical/Substance | − | − | − | − | − | x | − | x | − | − | x* | x* | Kovarich et al. (2020) |
| | Radiation | x | x | − | − | x | − | − | x | − | − | x* | x* | CNS (2020) |
| | Structure hazards | x | − | x | x | x | x | − | x | − | − | x* | − | eMARS (2020) |
| Transportational | Aviation | x | − | x | − | x | x | − | x | − | − | x | x | ICAO (2020) |
| | Railway | x | − | x | x | x | x | − | x | − | − | x | x | ERAIL (2020) |
| | Maritime | x | − | x | x | x | x | − | x | − | − | x | x | IMO (2020) |
| | Traffic | − | − | x | − | − | − | − | x | − | − | x* | − | ITF (2020) |
| **Social Media** | | x* | − | x* | x* | x* | x* | − | x* | x | x | x | x | |

## 4   Opportunities of Social Media Data

From the results of the survey shown in Table 3, the following six major opportunities of social media data for incident databases can be inferred: (1) More precise assessment of the impact of an incident across the board of disaster types, (2) generation of narratives or short descriptions, especially for droughts, geophysical incidents, and floods, (3) strengthening of the acquisition processes that already involve citizens, which is the case for more than half of the natural disasters surveyed, (4) support of weakly institutionalized regions and extension of surveillance areas, (5) narrow the areas for spatially constrained surveillance, and (6) noticing trigger events and enabling periodical surveillance.

The first opportunity—to more precisely assess the impact of an incident—can be inferred from the difference between existing impact data (in 93% of all incidents) and complete impact data (in 66% of all incidents). This large gap between existing and complete data suggests that this data is frequently required but difficult to acquire. Determining the impact of an incident mostly happens by local observation. Thus, impact data is more difficult to obtain when incidents are quantitatively surveilled, which is common for natural disasters, while man-made incidents are often qualitatively recorded in written reports with a focus on impact assessment. Deriving reliable impact data from quantitative data is difficult, even more so for subtle incident types like droughts (Enenkel et al., 2020). The required observations are frequently shared on social media as images and discussions, as well as personal or third-party observations, creating an opportunity to acquire the missing information.

The second opportunity—to generate narratives or short descriptions—can also be inferred from the difference between existing (in 85% of all incidents) and complete data (in 70% of all incidents). Narratives are short summaries of the episode, and despite their frequent existence, generating narratives by incorporating social media data as additional source might serve to reduce the effort required from the experts that create them manually. The incident types that would profit most from narrative generation are geophysical incidents such as earthquakes (showcased by Rudra et al. (2016)), droughts, and floods (showcased by Shapira et al. (2017)). Social media may be used as a basis to generate narratives due to the typically high velocity of information dissemination.

The third opportunity—to recruit citizen volunteers—can be inferred from the surveyed sources (surveillance systems, experts, and citizen observations), which describe how the incidents were originally reported. Citizen observations constitute a substantial part of the acquisition process in 75% of the surveyed natural disasters, particularly when surveillance sources are scarce. Examples are the severe weather reports collected by NOAA and ESWD, NASA's crowdsourcing of landslide data (Juang et al., 2019), and the flood impact observations within PetaJakarta (Ogie et al., 2019). Citizen observations are never noted as a source for man-made incidents. Here, all entries are from involved parties, like train operators or plane engineers. It is conceivable to involve citizens in the acquisition of traffic, industrial, and extreme transportation incidents. The data acquisition for these incident types is often limited by the ability to recruit volunteers or crowdworkers, where the recruitment relies on citizen initiative, and participation demands training with specialized tools, websites, and workflows. Social media platforms simplify contacting potential volunteers and may alleviate the burden of learning specialized tools (Mehta et al., 2017).

The fourth opportunity—to support weakly institutionalized regions and extend surveillance areas—can be inferred from the frequency of incomplete records (77% of the total) of incident types requiring dense spatial surveillance. If a dense surveillance infrastructure is necessary for a large area, weakly institutionalized regions that are limited by personal, technical, jurisdictional, or financial issues fall behind. The practical consequence is a frequent west-world bias in data collections as shown by Lorini et al. (2020) for the uneven coverage of floods on Wikipedia. Similar problems exist for meteorological surveillance in sparsely settled or poor areas without a network of weather stations, for wildfire monitoring without terrestrial camera networks, for droughts without precipitation monitoring, and, to a certain degree, for disease outbreaks.[2] Social media can aid the acquisition process of incidents requiring spatially dense surveillance infrastructure to a certain degree, since it is also used in sparsely settled and less developed regions.

---

[2] A study by Wang et al. (2020) suggests that Covid-19 could have been detected via social media weeks before the acknowledgment by officials institutions.

The fifth opportunity—To narrow the areas for spatially constrained surveillance—-can be inferred from the frequency of incomplete records (43% of the total) of incident types requiring constrained spatial surveillance which typically occur in a large, high-risk areas. Examples are fire watches, tornado spotting, and substance pollution and structural hazard monitoring. Surveying these risk areas requires many distributed, mostly human spotters (Brotzge, J. and Donner, W. , 2013), which may be found ubiquitously on social media. In practice, these signals can be used to trigger detailed surveillance systems, as showcased by Rashid et al. (2020), who used social media to route drones for disaster surveillance.

The sixth opportunity—To notice trigger events and start periodical surveillance—can be inferred from the frequency of incomplete records (33% of the total) of incident types requiring periodical temporal surveillance. These events typically require manually recognized trigger events to initiate and guide detailed surveillance, like wildfires, floods, and diseases. Social media can help to detect these trigger events through shared first- and third-party observations. Similarly, social media can assist the 22% of periodically surveyed incidents with potentially long intervals in their periodical surveillance, like space-based earth observation or scheduled contamination tests.

The survey also reveals two minor opportunities: establishing incident causality and assessing the response, recovery, and mitigation efforts, although we are cautious to point to social media data as a potential solution without significant prior academic effort. Firstly, the causal relations in the databases mostly mention the main cause, e.g., if a flood has been caused by a hurricane. This knowledge is naturally incomplete if the cause is the normal operation of earth systems, for example, for earthquakes. However, causal inference through social media data is sought after for sub-events (Chen and Terejanu, 2018), like road-blocks caused by a storm, which is in this granularity not captured by our survey. Secondly, assessment of the response, recovery, and mitigation efforts are frequent for man-made disasters, where humans have more agency in prevention, but rare for natural ones, where assessments are often only created for significant incidents or in annual reports. There is an apparent value in generating assessments for individual natural incidents, but it is not yet clear how.

Furthermore, the survey hints at areas without an apparent need to utilize social media data. Metadata are largely (93%) existent and complete if the incident is known. A similar argument can be made for sensory information, however, there is pioneering work on crowdsourcing sensory information from citizen observations, for example, inferring hail diameters or flood levels from posted images (Assumpção et al., 2018). Besides, there is no apparent need to use social media data to increase the spatial resolution if the incidents are surveyed globally through earth observation techniques and have reliable forecasting models, for example, in the case of hurricanes. Also, there is no apparent need for incidents with static or strictly tracked constrained extent, like the surveillance of regionally limited incidents, earthquakes and volcano eruptions, and the scrutiny of incidents that have reliable surveillance systems in place, like transportational incidents.

In the opportunities derived above, the task in focus is often to be the evaluation of citizen-provided information by authorities. The prevalent communication type of the used social media content might indeed be related to citizens' self-coordination (Reuter and Kaufhold, 2017) and information exchange. However, a more active approach of involvement (e.g., digital volunteers (Starbird and Palen, 2011)), official information distribution (Plotnick and Hiltz, 2016) and participation (i.e., a bi-directional communication between involved actors and affected citizens) has the potential to support information gathering for resource planning and local forecasting activities.

## 5  Risks of Social Media Data

When modeling the acquisition process for incident information as a process that derives relevant data from a given source, then the quality of this process can be judged in terms of the well-known measures precision (correctness and reliability) and recall (completeness) of the data derived. In general, the risks of using social media as a source of information are founded in the fact that, unlike the other sources, social media is not inherently reliable. At present, using social media requires a trade-off between precision and recall, since no "perfect" solution for its analysis is available.

The survey results in Table 1 show that the traditional sources of information (namely sensor-based surveillance systems, experts, and citizen volunteers) often lack recall, but are optimized for precision through engineering, education, and expert scrutiny. Our survey results in Section 4 show that social media has the potential to increase the overall recall of all other information sources combined, though. But as a kind of "passive" crowdsourcing, the precision of social media must be expected to be significantly lower than, for example, that of "active" crowdsourcing using citizen volunteers: Social media users are anonymous and only a posteriori quality control is possible, whereas, as per Wiggins et al. (2011),[3] quality assurance strategies for traditional citizen science projects can be applied before data collection (e.g., training, vetting, and testing citizens), during data collection (e.g., specialized tooling and evidence requirements), after data collection (e.g., expert reviews, statistical analysis of the participants and the data), and by contacting the citizen volunteers.

Several studies outline the general data quality issues of social media for disaster-related use cases, such as its credibility (Castillo et al., 2011) and trustworthiness (Nurse et al., 2011; Tapia and Moore, 2014). Stieglitz et al. (2018) refer to the big data quality criteria (the "Big V"s) for social media data, and in particular highlight variety and veracity as problems. Here, as per Lukoianova and Rubin's 2013 definition, veracity is understood as objectivity, truthfulness, and credibility. However, no general guidelines or best practices for data quality assurance have been established, yet. Thus, the precision and the recall of social media as a source of incident information must be traded off, and it remains to be seen whether precision can be maximized without sacrificing so much recall that this source of information practically adds nothing new to the traditional ones combined. And this may depend on the type of incident or disaster at hand, its characteristics, and the opportunities pursued.

### 5.1  Tradeoff when using Social Media

With the notion of incident data acquisition as a process that produces data from the given sources within spatio-temporal constraints in mind, assessing the risks of social media is equivalent to studying the effects of including versus excluding social media data as an information source. Generally, excluding social media data leads to the information gaps revealed by our survey and including social media data leads the issues studied by the aforementioned related work relating to objectivity, truthfulness, credibility, and trust. Although this general notion is macroscopically correct and useful, the detailed trade-off in different scenarios is scarcely explored and varies substantially. We illustrate the trade-off of acquiring incident information from social media with implications of our survey and clues from the related work, categorized within the scope of each opportunity, as summarized in Table 4.

---

[3]A similar analysis is presented by Meek et al. (2014) for other kinds of crowdsourcing projects.

**Table 4.** Overview of the trade-off when using social media data to gather incident information by application scenario. The risks of **excluding social media** as information source are due to lower recall, and those of **including social media** means are due to lower precision.

| | Opportunity | Excluding Social Media | Including Social Media |
|---|---|---|---|
| (1) | Impact assessment and model verification | Susceptible to model imprecision and bias. Impact can not be estimated if data are missing. Field surveys are expensive. | Impact is imprecise if source is too noisy. Impact is imprecise if metadata is missing. |
| (2) | Narrative generation | Narrative lacks critical facts. Narrative is missing. | Narrative is dominated by trivia and sentiment. Narrative is imprecise if facts change rapidly. Narrative is imprecise in cross-lingual settings |
| (3) | Volunteer recruitment | Few volunteers if task requires complex tooling. Few volunteers if task requires training. Few volunteers if task is time consuming. | Volunteers may be biased (opportunism, misuse). Data acquisition and use has unsolved ethical issues. (Long-term) availability is unclear. Depends on availability of motivated users |
| (4) | Support weakly institutionalized areas | Incident is missed. Imprecision if sensors are sparse. | Participatory Inequality Susceptible to outliers. |
| (5) | Narrow surveillance area | Incident is missed. | Area is imprecise if source is not geotagged. Susceptible to outliers. |
| (6) | Report triggers | Incident is missed. | Susceptible to outliers. Errors are expensive. |

Within the first opportunity—to more precisely assess the impact of an incident—excluding social media leads to missing information, possibly imprecise estimates, and expensive excursions. Impact assessment requires substantial local observation, so the quality of the assessment often depends on the number of trained observers and the existing support infrastructure. Since both are expensive and often unavailable, estimations and modeling are frequently used tools, which naturally introduce imprecision and are susceptible to biases. These effects could be mitigated by including local observations shared through social media. The risks of including social media data are that of introducing other kinds of imprecision. Social media contains exclusively qualitative data,[4] essential metadata is often coarse (like city-level geolocation instead of coordinates), and, data, such as shared imagery, is often too complex for impact models (Nguyen et al., 2017; Ogie et al., 2019; Fang et al., 2019). Consequently, in the related work, impact assessments derived from social media are often highly imprecise: for instance, Hao and Wang (2020) resort to 500-meter cells for flood mapping, and Fohringer et al. (2015) report flood inundation mapping errors at the scale of decimeters.

Within the second opportunity—to generate narratives or short descriptions—excluding social media data leads to missing narratives or narratives that miss the critical, but not previously observed facts. These effects can be mitigated by deriving the necessary facts or narrative steps from information shared on social media. The risks of including social media data are distortions towards trivia, imprecision in evolving situations, and imprecision in cross-lingual settings. Since sharing sentiment and discussion is one of the primary uses of social media in a crisis (Palen and Liu, 2007), factual information is easily drowned and narrative generation may become biased towards trivia and sentiment, as noted by Alam et al. (2020). Similarly, Rudra et al. (2018) notes that extracting facts from social media relies on semi-automated filtering, and, although these methods achieve

---

[4]Crooks et al. (2013) refer to social media data as "Ambient Geographic Information", in contrast to "Volunteered Geographic Information".

accuracies past the 90%, critical facts can still be lost. They also conclude that, for example, impact facts change as an incident progresses and users may not share the most recent information at a given time, so the chronological order on social media may not correspond to the actual order of events, giving rise to conflicting points of view. Rudra et al. (2018) also demonstrate issues in cross-lingual settings, since social media discussions on a single topic feature multiple languages and code-switching. Besides the unsolved algorithmic challenges, this also poses challenges to human assessors. Concluding, Aslam et al. (2015) notes as a result of the TREC-2015 temporal summarization track that automated systems "either had a fairly high precision or novelty with topic coverage, [...] and it appears that attaining high precision is more difficult than achieving recall".

Within the third opportunity—to recruit citizen volunteers—excluding social media data leads to less volunteers due to the barriers of entry introduced by lack of awareness, complex tooling, training requirements, and the required time investment, which are partially related to the quality assurance practices. These effects can partially be mitigated by recruiting volunteers for simple tasks through social media. The risks of recruiting volunteers through social media are lack of motivation and biases of the users. Ogie et al. (2018) show that the likelihood of participating and the likelihood of contributing valuable information is lower for ordinary citizens than for response personnel. They conclude that the quality of crowdsourcing via social media heavily depends on the user's perception of the value of their data and the user's exposure to the incident (Ogie et al., 2019). In a review of 169 studies of passive crowdsourcing in environmental research, Ghermandi and Sinclair (2019) additionally conclude ethics and long-term availability as issues with volunteers on social media.

Within the fourth opportunity—to support weakly institutionalized regions—excluding social media data leads to missing or imprecise incident data, particularly when sensors or experts are sparse and measures can only be triangulated or estimated. These effects can be mitigated by including local observations shared through social media. The risks of including social media data are imprecision due to participatory inequality, and, as a consequence, susceptibility to outliers. Weakly institutionalized regions are highly susceptible to participatory inequality (Ogie et al., 2019) in that fewer but more educated and motivated users dominate social media in these regions, distorting the picture. Xiao et al. (2015) and Wang et al. (2019) confirm that users from socially vulnerable areas share less information via social media and that social media data may not reveal the true picture due to the uneven access to social media and heterogeneous motivations in social media usage. As a consequence, the better-situated areas within weakly institutionalized regions may appear more vulnerable or more affected by incidents.

Within the fifth opportunity—to narrow the areas for spatially constrained surveillance—-excluding social media data leads to missing incident data. These effects can be mitigated by determining areas for thorough surveillance based on observations shared through social media. The risks of including social media data are imprecision due to inconsistent metadata and susceptibility to outliers. Metadata like geotags are essential when drawing geographical conclusions from social media data, like the area to survey. However, geotags are often optional or coarse-grained, such as city-level, so precise information must be predicted or estimated, inevitably introducing imprecision. If sufficient observations are shared, the area for spatially constrained surveillance can be triangulated to mitigate the imprecision (Senaratne et al., 2017). Without sufficient observations, the conclusions are susceptible to outliers, particularly if the endangered area is large and uninhabited so that observations naturally become sparse. These outliers occur either when coordinates are wrongly estimated, or in the reported cases of misuse like phishing (Verma et al., 2018), fake news (Zhang and Ghorbani, 2020), and rumors or exaggerations (Mondal et al., 2018).

Within the sixth opportunity—to notice trigger events and start periodical surveillance—the risks of excluding or including social media data align with the fifth opportunity. Noticing trigger events is susceptible to outliers in low activity regions and for low impact incidents where online discussion is limited. Wrongly noticing triggers for incident types like wildfires, floods, and diseases may lead to expensive surveillance campaigns, blocking resources, and harming citizen trust in institutions.

## 6 Conclusions

This work attempts to answer which role social media data can play in disaster management by systematically surveying the currently available data in 37 disaster and incident-databases, assessing the missing and sought-after information, pointing out the opportunities of information spread via social media to fill these gaps, and ponder the risks of including social media data in the traditional acquisition process. The identified gaps hint at six primary opportunities: impact assessment and verification of model predictions, narrative generation, enabling enhanced citizen involvement, supporting weakly institutionalized areas, narrowing surveillance areas, and reporting triggers for periodical surveillance. Additionally, we point to potential opportunities warranting further research: determining causality between incidents and sub-events, and generating assessments about the response, recovery, and mitigation efforts. Given proper awareness of the risks, seizing the determined opportunities and including social media-based citizen observations in incident data acquisition can greatly improve our ability to analyze, cope with, and mitigate future disasters. However, we conclude that social media should not be included undifferentiated, but as tool to mitigate the weaknesses of traditional sources for specific data needs in specific incidents and application scenarios.

### 6.1 Limitations

In favor of following a reproducible and data-driven approach to surveying, we do not consider information that may be needed but is never contained in any of the databases. This also means that we do not suggest to limit innovation or research when rejecting use cases like earthquake detection or metadata extraction. There may be novel uses for social media data which are not revealed by our survey. Additionally, our analysis does not consider the uses of social media analysis to reduce detection times and applications that use social media to retrieve other sources, like shared news articles. Note that we limited our conclusions about traffic incidents due to the limited data in IRTAD and that we mostly ignored uncommon and unforeseen events because of the naturally limited data to survey.

*Author contributions.* MW, JK conducted the database survey and the survey of the related work. MW provided the inital draft with contributions from JK. HS provided the idea and initial draft of Section 5. MP, FK, and BS revised the manuscript. All authors contributed to the analysis.

*Competing interests.* No competing interests are present.

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
