# Peer review of "Opportunities and Risks of Disaster Data from Social Media: A Systematic Review of Incident Information"

_Natural Hazards and Earth System Sciences, 2020_

## Referee Comment (RC1) · Chiara Arrighi (Referee) · 4 Aug 2020

The manuscript "Opportunities and Risks of Disaster Data from Social Media: A Systematic Review of Incident Information" reviews 37 incident databases (both expert-based and citizen-based) to highlight the most appropriate use and uncertainties of data from social media in disaster management. The manusscript is well written and structured and summarizes very effectively the information included in the accident databases according to a rational taxonomy. The results show that the opportunities of social media are overcome by uncertainty, especially from the scientific perspective.

My suggestion is to follow the six primary opportunities identified in section 4 when

describing the uncertainties in sect. 4.2. For example, point (2) is the generation of narratives and short description. Here the limitation might be that the narrative can be misleading, e.g. suggesting wrong or unwise behaviour. For floods a classic example is when people report selfies on flooded bridges, or drive a car in flooded roads. There should be a point-by-point description of the potential limitations after the general description of error types, or the text could be replaced by a table "opportunities-uncertainties". Given the above, I would also add a bit more discussion in the conclusive section.

---

## Referee Comment (RC2) · Anonymous Referee #2 · 18 Sep 2020

The manuscript "Opportunities and Risks of Disaster Data from Social Media: A Systematic Review of Incident Information" aims at understanding whether, and under which circumstances, social media can supply relevant information (both qualitatively and quantitatively) in risk analyses. A very large database of social media information is considered, which includes 37 large disaster and incident databases covering 27 incident types, assessing the quality of data and the missing information according to a stadardized taxonomy. The manuscript is well written and structured and I encourage its pubblication in NHESS. My only suggestion is to review the discussion sections by adding a more detailed list of the uncertainties and limitation of this type of data, maybe using examples taken from the events. I think this point of view is quite innovative and

is worth analysing it very carefully.

---

## Author Comment (AC1) · 23 Oct 2020

**Answers to the Reviewers' Comments**

October 23, 2020

We wish to thank the reviewers and editors for their helpful comments, which we carefully considered and incorporated in the revised version. Based on all reviewers' comments, we will attempt the following additions to Section 4.2, so it will be more specific, sensible, and easier to follow:

- We will add/extend the description of the uncertainty types to specify what they entail, and we will provide evidence of the uncertainties for different types of incidents. A table for illustration will be considered.

- We will extend the section by going through all opportunities and highlight the uncertainties affecting them, and to what degree.

**Reviewer 1**

**Comment 1.** *My suggestion is to follow the six primary opportunities identified in section 4 when describing the uncertainties in sect. 4.2. For example, point (2) is the generation of narratives and short description. Here the limitation might be that the narrative can be misleading, e.g. suggesting wrong or unwise behaviour. For floods*

*a classic example is when people report selfies on flooded bridges, or drive a car in flooded roads.*

**Answer 1.** This is a nice suggestion. Our initial intention was to focus on deriving the opportunities in a quantitative, data-driven fashion. We assumed that the encountered uncertainties heavily depend on the type of disaster—we can find very different incarnations of Type I and Type II errors in floods and in pandemics—as well as the opportunity. We thus settled on providing a general overview, whereas compiling the full scope of incarnations seemed unrealistic. However, following the six opportunities, and providing evidence for the existence of different forms of uncertainties seems indeed necessary and manageable.

**Comment 2.** *There should be a point-by-point description of the potential limitations after the general description of error types, or the text could be replaced by a table "opportunities-uncertainties".*

**Answer 2.** Following up on the previous point, we agree that evidence for the different uncertainties is necessary. It can be expected that all types of uncertainties mentioned will be a problem for all opportunities and require a different reaction based on the disaster type. Compiling a table may be possible, but we have to carefully inspect if contrasting opportunities and uncertainties in this form will tunr out to be suitable, given the complexity of the problem.

**Comment 3.** *Given the above, I would also add a bit more discussion in the conclusive section.*

**Answer 3.** We will review the discussion section, consolidate the findings from the previous sections, and add new evidence we might find during the extension of Section 4.2.

---

## Author Comment (AC2) · 23 Oct 2020

**Answers to the Reviewers' Comments**

October 23, 2020

We wish to thank the reviewers and editors for their helpful comments, which we carefully considered and incorporated in the revised version. Based on all reviewers' comments, we will attempt the following additions to Section 4.2, so it will be more specific, sensible, and easier to follow:

- We will add/extend the description of the uncertainty types to specify what they entail, and we will provide evidence of the uncertainties for different types of incidents. A table for illustration will be considered.

- We will extend the section by going through all opportunities and highlight the uncertainties affecting them, and to what degree.

**Reviewer 2**

**Comment 1.** *My only suggestion is to review the discussion sections by adding a more detailed list of the uncertainties and limitations of this type of data, maybe using examples taken from the events. I think this point of view is quite innovative and is worth analyzing it very carefully.*

**Answer 1.** We assume that all types of uncertainties will occur in social media data, independently of the opportunity and in a different incarnation depending on the incident type. Therefore, we settled on giving a more general overview of what researchers and practitioners have to inspect depending on their specific question. We agree that the uncertainties section needs both, more explanation and specific evidence.

---

## Author Response (AR1)

**Answers to the Reviewers' Comments**

December 23, 2020

We wish to thank the reviewers and editors for their helpful comments, which we carefully considered and incorporated in the revised version. Based on all reviewers' comments, we completely revised the previous Section 4.2 (now Section 5). This new section discusses the risks of social media as a source of incident information by highlighting both, the conclusion from the survey of the incident databases and the findings in the related work, to more precisely describe the tradeoff to be expected when including vs. excluding social media data. In detail:

- We discuss the issues of social media data in disasters in general and for incident data collection specifically in Section 5.

- We introduced a formalism for the data acquisition process, which describes social media data as an additional source.

- Following this formalism, we describe the expected tradeoff when including social media data as an increase in recall of the process against a loss in precision.

- We show evidence for the tradeoff from our survey and the related work within the scope of each opportunity. The resulting detailed and differentiated discussion forms Section 5.1 and Table 4.

**Response to Reviewer 1**

**Comment 1.** *My suggestion is to follow the six primary opportunities identified in section 4 when describing the uncertainties in sect. 4.2. For example, point (2) is the generation of narratives and short description. Here the limitation might be that the narrative can be misleading, e.g. suggesting wrong or unwise behaviour. For floods a classic example is when people report selfies on flooded bridges, or drive a car in flooded roads.*

**Answer 1.** We followed this suggestion. Section 5.1. now follows the six opportunities, similarly to Section 4, and describes both, the risk of including social media data (i.e. limitations) and the risks of not doing so, according to the evidence we found in the corresponding related work.

**Comment 2.** *There should be a point-by-point description of the potential limitations after the general description of error types, or the text could be replaced by a table "opportunities-uncertainties".*

**Answer 2.** We added Table 4 to the manuscript. It contrasts the tradeoff for each opportunity in more detail. We found that the notion of error types was too general to discuss the risks of social media data on a per-opportunity basis. We condensed the general discussion (previously Section 4.2) in Section 5.

**Comment 3.** *Given the above, I would also add a bit more discussion in the conclusive section.*

**Answer 3.** We expanded the relevant parts of the discussion in Section 5 extensively to cover this point. We added a concluding remark in the conclusion reflecting the findings of the discussion from Section 5.

**Response to Reviewer 2**

**Comment 4.** *My only suggestion is to review the discussion sections by adding a more detailed list of the uncertainties and limitations of this type of data, maybe using examples taken from the events. I think this point of view is quite innovative and is worth analyzing it very carefully.*

**Answer 4.** We found that the notion of error types was too general to discuss the risks of social media data on a per-opportunity basis. We expanded the discussion of the risks and tradeoffs of social media data in greater detail with findings from the related work. We think this illustrates the problem better than selected Examples.

**List of Changes**

We thoroughly revised the writing of the manuscript throughout to improve readability and ease of understanding, hence providing a track-changes file provides little value. Instead, we list all major changes in the manuscript here.

**0.1 Abstract**

10 Renamed opportunity 3 to "recruiting citizen volunteers" to better reflect the intended meaning.

**0.2 Section 1: Introduction**

37ff Rephrased the research question (2) to better reflect the intended meaning and research question (3) to reflect our extended notion of the risks.

49ff Reworked the description of contribution 3 to reflect our extended notion of the risks.

**0.3 Section 4: Opportunities of Social Media Data**

- Added additional examples from the related work to provide a more comprehensive perspective.

- Moved Section 4.2 (Uncertainties) to Section 5.

- Moved Section 4.3 (Limitations) to Section 6 (Conclusion) to better reflect it's purpose of putting our results in perspective.

168ff Rephrased the description of the 3rd opportunity to better reflect the intended meaning.

**0.4  Section 5: Opportunities of Social Media Data**

- New Section.

- Extended the discussion of the risks of social media (previous 4.2, see responses to the reviewers)

- Section 5.0. provides a general perspective on the risks of using social media data.

- Section 5.1 discusses the tradeoff of including vs. excluding social media data individually for each of the opportunities from Section 4.

**0.5  Section 6: Conclusion**

319ff  Added a remark to also conclude our findings from the revision of Section 5.

---

## Referee Report (RR1)

Authors have carefully addressed the comments raised by the reviewers. The manuscript is now clearer and the section 5 is well organized. I thus recommend to accept the manuscript.